# Markov Information Bottleneck to Improve Information Flow in Stochastic Neural Networks

**DOI:** 10.3390/e21100976

**Published:** 2019-10-06

**Authors:** Thanh Tang Nguyen, Jaesik Choi

**Affiliations:** 1Applied Artificial Intelligence Institute, Deakin University, Geelong VIC 3220, Australia; 2Graduate School of Artificial Intelligence, Korea Advanced Institute of Science and Technology, Daejeon 34141, Korea

**Keywords:** information bottleneck, stochastic neural networks, variational inference, machine learning

## Abstract

While rate distortion theory compresses data under a distortion constraint, information bottleneck (IB) generalizes rate distortion theory to learning problems by replacing a distortion constraint with a constraint of relevant information. In this work, we further extend IB to multiple Markov bottlenecks (i.e., latent variables that form a Markov chain), namely Markov information bottleneck (MIB), which particularly fits better in the context of stochastic neural networks (SNNs) than the original IB. We show that Markov bottlenecks cannot simultaneously achieve their information optimality in a non-collapse MIB, and thus devise an optimality compromise. With MIB, we take the novel perspective that each layer of an SNN is a bottleneck whose learning goal is to encode relevant information in a compressed form from the data. The inference from a hidden layer to the output layer is then interpreted as a variational approximation to the layer’s decoding of relevant information in the MIB. As a consequence of this perspective, the maximum likelihood estimate (MLE) principle in the context of SNNs becomes a special case of the variational MIB. We show that, compared to MLE, the variational MIB can encourage better information flow in SNNs in both principle and practice, and empirically improve performance in classification, adversarial robustness, and multi-modal learning in MNIST.

## 1. Introduction

The information bottleneck (IB) principle [1] extracts relevant information about a target variable *Y* from an input variable *X* via a *single* bottleneck variable *Z*. In particular, it constructs a *bottleneck* variable Z=Z(X) that is a *compressed* version of *X* but preserves as much *relevant* information in *X* about *Y* as possible. This principle of introducing relevant information under compression finds vast applications in clustering problems [2], neural network compression [3], disentanglement learning [4,5,6,7], and reinforcement learning [8,9]. In addition, there have been many variants of the original IB principle, such as multivariate IB [10], Gaussian IB [11], meta-Gaussian IB [12], deterministic IB [13], and variational IB [14]. Despite these vast applications and variants of IB, alongside the theoretical analysis of the IB principle in neural networks [15,16], the context of stochastic neural networks in which mutual information can be most naturally well-defined [17] has not been sufficiently studied from the IB insight. In this work, we are particularly interested in this context in which multiple stochastic variables are constructed for representation in the form of a Markov chain.

Stochastic neural networks (SNNs) are a general class of neural networks with stochastic neurons in the computation graph. There has been an active line of research in SNNs, including restricted Boltzmann machines (RBMs) [18], deep belief networks (DBNs) [19], sigmoid belief networks (SBNs) [20], and stochastic feed-forward neural networks (SFFNs) [21]. One of the advantages of SNNs is that they can induce rich multi-modal distributions in the output space [20] and enable exploration in reinforcement learning [22]. For learning SNNs (and deep neural networks in general), the maximum likelihood estimate (MLE) principle (in its various forms, such as maximum log-likelihood or Kullback–Leibler divergence) has generally been a de-facto standard. The MLE principle maximizes the likelihood of the model for observing the entire training data. However, this principle is generic and not specially tailored to the hierarchical structure of neural networks. Particularly, MLE treats the entire neural network as a whole without considering the explicit contribution of its hidden layers to model learning. As a result, the information contained within the hidden structure may not be adequately modified to capture the data regularities reflecting a target variable. Thus, it is reasonable to ask if the MLE principle effectively and sufficiently exploits a neural network’s representative power, and whether there is a better alternative.

**Contributions**. In this paper, (i) we propose Markov information bottleneck (MIB), a variant of the IB principle for multiple Markov bottlenecks that directly offers an alternative learning principle for SNNs. In MIB, there are multiple bottleneck variables (as opposed to one single bottleneck variable in the original IB) that form a Markov chain. These multiple Markov bottlenecks sequentially extract relevant information for a learning task. From the perspective of MIB, each layer of an SNN is a bottleneck whose information is encoded from the data via the network parameters connecting the layer to the data layer. (ii) We show that in a non-collapse MIB, the information optimality is not simultaneously achievable for all bottlenecks; thus, an optimality compromise is devised. (iii) When applied to SNNs for a learning task, we interpret the inference from a hidden layer to the output layer in SNNs as a variational approximation to that layer’s intractable decoding of relevant information. Consequently, the variational MIB in SNNs generalizes the MLE principle. We demonstrate via a simple analytical argument and synthetic experiment that MLE is unable to learn a good information representation, while the variational MIB can. (iv) We then empirically show that MIB improves the performance in classification, adversarial robust learning, and multi-modal learning in the standard hand-digit recognition data MNIST [23]. This work is an extended version of our preprint [24] and the first author’s Master thesis [25].

## 2. Related Work

There have been many extensions of the original IB framework [1]. One natural consideration is to extend it to continuous variables, yet under special settings where the optimal information representation is analytic [11,12]. Another direction uses alternative measures for compression and/or relevance in IB [13]. Since the optimal information representation in IB is tractable only in limited settings such as discrete variables [1], Gaussian variables [11], and meta-Gaussian variables [12], scaling the IB solution using neural networks and variational inference is a very successful extension [14]. The closest extension to our MIB is multivariate IB [10], in which they define multi-information to capture the dependence among the elements of a multivariate variable. However, in MIB, we do not focus on capturing such multi-information but rather the optimal information sequentially processed by a Markov chain of (possibly multivariate) bottleneck variables.

The line of work applying the IB principle to learn information representation in neural networks is also relevant to our approach. For example, Reference [15] proposes the use of the mutual information of a hidden layer with the input layer and the output layer to quantify the performance of neural networks. However, it is not clear as to how the IB information optimality changes in multiple bottlenecks in a neural network and how we can approximate the IB solutions in this high-dimensional context. In addition, MLE is a standard learning principle for neural networks. It has been shown that the IB principle is mathematically equivalent to the MLE principle in the multinomial mixture model for the clustering problem when the input distribution *X* is uniform or has a large sample size [26]. However, it is also not clear how these two principles are related to each other in the context of neural networks. Moreover, regarding the feasibility of the IB principle for representation learning in neural networks, Reference [17] analyzes two critical issues of mutual information that representation learning might suffer from: indefinite in deterministic encoding, and invariant under bijective transformations. These are inherent properties of mutual information which are also studied in, for example, [7,27,28]. In MIB, we share with [17] the same insight in these caveats by considering only the scenario where mutual information is well defined. This also explains our rationale in applying MIB to stochastic neural networks.

Deep learning compression schemes [3,29] loosely bear some similarity with our work. Both of the directions aim for a more compressed and useful neural networks for given tasks. The critical distinction is that deep learning compression schemes attempt to produce a smaller-sized neural network with similar performance of a larger one so that the network can be efficiently deployed in small devices such as mobile phones. This task therefore involves size-reduction techniques such as neural network pruning, low-rank factorization, transferred convolution filters and knowledge distillation [29]. On the other hand, our work asks an important representation learning question that given a neural network, what learning principles are the best we can do to improve the information content learned from the data for a given task? In this work, we attempt to address this question via the perspective that a neural network is a set of stochastic variables that sequentially encode information into its layers. We then explicitly improve the information flow (in the sense of more compressed but relevant information) for each layer via our introduced Markov Information Bottleneck framework.

## 3. Preliminaries

### 3.1. Notations

We denote random variables (RVs) by capital letters (e.g., *X*), and their specific realization value by the corresponding lowercase letter (e.g., *x*). We write X⊥Y (respectively, 
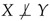
) to indicate that *X* and *Y* are independent (respectively, not independent). We denote a Markov chain by Y→X→Z, that is, *Y* and *Z* are conditionally independent given *X*, or Y⊥Z|X. We use the integral notation when taking expectation (e.g., ∫p(x)f(x)dx) over the distribution of a random variable regardless of whether the variable is discrete or continuous. We also adopt the following conventions from [27] for defining entropy (denoted by *H*), mutual information (denoted by *I*), and Kullback–Leibler (KL) divergence (denoted by $D_{KL}$): $0\log\frac{0}{0}=0,0\log\frac{0}{q}=0,p\log\frac{p}{0}=\infty$.

### 3.2. Information Bottleneck

Given a (possibly unknown) data joint distribution p(X,Y), the IB framework constructs a *bottleneck* variable Z=Z(X) that is a *compressed* version of *X* but preserves as much *relevant* information in *X* about *Y* as possible. The compression of the representation *Z* is quantized by I(Z;X), the mutual information of *Z* and *X*. The relevance in *Z*, the amount of information *Z* contains about *Y*, is specified by I(Z;Y). The optimal representation *Z* satisfying a certain compression–relevance trade-off constraint is then determined via minimization of the following Lagrangian $L_{IB}\left[p\left(z|x\right)\right]=I\left(Z;X\right)-\beta I\left(Z;Y\right)$, where β is a positive Lagrangian multiplier that controls the trade-off. Due to the convexity of Lagrangian and constrained conditions with respect to the encoders {p(z|x)}, the Karush–Kuhn–Tucker (KKT) conditions for this constrained minimization problem become the sufficient and necessary conditions for finding the optimal encoders {p(z|x)}. By solving the KKT conditions, we can obtain the optimal encoders which can be expressed in an energy-based form as the following:(1)$$\begin{array}{c} \underset{p(z|x)}{arg min}L_{IB}\left[p\left(z|x\right)\right]\propto p\left(z\right)\exp\left(-\beta D_{KL}\left[p(Y|x)∥p(Y|z)\right]\right), \end{array}$$
where p(z)=∫p(z|x)p(x)dx.

## 4. Markov Information Bottleneck

Given a data joint distribution p(X,Y) which is possibly only observed via a set of i.i.d. samples $S=\{{\left(x_{i},y_{i}\right)}_{i=1}^{N}\}$, an information representation *Z* for p(X,Y) is said to be good if it encodes sufficient relevant information in *X* about *Y* in a compressed manner. Ideally, *Z* summarizes only the relevant information in *X* about *Y* and discards all the irrelevant information; more formally, *Z* is a minimal sufficient statistic for *Y*. Such information representation is desirable because it can capture the regularities in the data and is helpful for generalization in learning problems [30,31]. Our main interest is in solving the optimal information representation for a latent variable *Z* that has Markov structure, that is, $Z=(Z_{1},Z_{2},\ldots ,Z_{L})$, where $Z_{1}\to Z_{2}\to \cdots \to Z_{L}$. The Markov structure is common in deep neural networks whose advantage is the powerful modeling capacity coming from multiple layers. In MIB, each encoder $p(z_{l+1}|x)$ relates the encoders of the previous bottlenecks in the Markov chain via Bayes’ rule:(2)$$\begin{array}{cc} p(z_{l+1}|x) & =\int p\left(z_{l+1},z_{1:l}|x\right)dz_{1:l}=\int \prod_{i=1}^{l+1}p\left(z_{i}|z_{i-1}\right)dz_{1:l},\forall 1\leq l\leq L-1, \end{array}$$
where $z_{1:l}:=\left(z_{1},\ldots ,z_{l}\right)$ and $z_{0}:=x$. In addition, each *encoder*
$p(z_{l}|x)$ corresponds to a unique decoder, namely *relevance decoder*, that decodes the relevant information in *x* about *y* from representation $z_{l}$:(3)$$\begin{array}{c} p\left(y|z_{l}\right)=\int p\left(x,y\right)\frac{p(z_{l}|x)}{p(z_{l})}dx. \end{array}$$

In MIB, we further introduce a surrogate target variable $\hat{Y}$ (for the target variable *Y*) into the Markov chain: $Y\to X\to Z_{l}\to Z_{l+1}\to \hat{Y}$ (Figure 1). The purpose of the surrogate target variable becomes clear in the section on variational MIB.

A trivial solution to the optimal information representation problem for *Z* is to apply the original IB principle for *Z* as a whole by computing the optimal IB solution in Equation (Equation 1). However, this solution ignores the Markov structure of *Z*. As a principled approach, leveraging the intrinsic structure of a problem can generally provide a new insight that goes beyond the limitation of the perspective that ignores such structure. Thus, in Markov information bottleneck (MIB), we explicitly leverage the Markov structure of *Z* to derive a principled and tractable approximate solution to the optimal information representation. We then empirically show that leveraging the intrinsic structure in the case of MIB is indeed beneficial for learning.

In MIB, we reframe the optimal information representation problem as multiple IB problems for each of the bottlenecks $Z_{l}$:(4)$$\begin{array}{c} \min_{p(z_{l}|x)}L_{l}\left[p\left(z_{l}|x\right)\right]:=\min_{p(z_{l}|x)}\left\{I\left(Z_{l};X\right)-\beta_{l}I\left(Z_{l};Y\right)\right\}, \end{array}$$
for all 1≤l≤L.

This extension is a natural approach for multiple bottlenecks because it aims for each bottleneck to achieve its own optimal information, and thus allows more relevant but compressed information to be encoded into *Z*. Another advantage is that we can leverage our understanding of the IB solution for each individual IB problem in Equation (Equation 4). Though this approach is promising and has good interpretation, there are two main challenges:The Markov structure among $Z_{l}$ prevents them from achieving their own information optimality simultaneously in non-trivial cases.The intractability of $p(y|z_{l})$ in Equation (Equation 3) and $p(z_{l}|x)$ in Equation (Equation 2) results in intractable mutual information in MIB.

In what follows, we formally establish and present the first challenge, the conflicting property of information optimality in Markov structure, in Theorem 1 followed by a simple compromise to overcome the information conflict. After that, we present variational MIB to address the second challenge.

Without loss of generality, we consider the case when L=2 (the result trivially generalizes to L>2). We first define the collapse mode of the representation *Z* to be the two extreme cases in which $Z_{2}$ either contains all the information in $Z_{1}$ about *X* or simply random noise:

**Definition** **1** (The *collapse* mode of MIB).
*$Z=(Z_{1},Z_{2})$ is said to be in the collapse mode if it satisfies either of the following two conditions:*
(1)
*$Z_{2}$ is a sufficient statistic of $Z_{1}$ for X and Y (i.e., $Y\to X\to Z_{2}\to Z_{1}$);*
(2)
*$Z_{2}$ is independent of $Z_{1}$.*



For example, if $Z_{2}=f\left(Z_{1}\right)$ where *f* is a deterministic bijection, $Z_{2}$ is a sufficient statistic for *X*. We then establish the conflicting property of information optimality in the Markov representation *Z* via the following theorem:
**Theorem** **1** (*Conflicting* *Markov* *Information* *Optimality*). *Given $X,Y,Z_{1},$ and $Z_{2}$ such that $Y\to X\to Z_{1}\to Z_{2}$ and H(Y|X)>0, consider two constrained minimization problems:*(5)$$\begin{array}{c} \underset{p(z_{l}|x)}{arg min}L_{l}\left[p\left(z_{l}|x\right)\right]:=\underset{p(z_{l}|x)}{arg min}\left\{I\left(Z_{l};X\right)-\beta_{l}I\left(Z_{l};Y\right)\right\},l\in \left\{1,2\right\}, \end{array}$$*where $0<\beta_{1}<\infty$, $0<\beta_{2}<\infty$, and $p\left(z_{2}|x\right)=\int p\left(z_{2}|z_{1}\right)p\left(z_{1}|x\right)dz_{1}$. Then, the following two statements are equivalent:*Z is not in the collapse mode;*The two optimal solutions to $L_{1}$ and $L_{2}$ in (Equation 5) are* conflicting, *that is, there is no single solution that minimizes $L_{1}$ and $L_{2}$ simultaneously.*

Theorem 1 suggests that the Markov information optimality conflicts for most cases of interest (e.g., stochastic neural networks, which we will present in detail in the next section). The values of $\beta_{1}$ and $\beta_{2}$ are important to control the ratio of the relevant information versus the irrelevant one presented in the bottlenecks. These values also determine the conflictability of multiple bottlenecks on the edge cases. Recall by the data processing inequality (DPI) [27] that for Y→X→Z, we have 0≤I(Z;X)≤H(X) and 0≤I(Z;Y)≤I(X;Y). If $\beta_{1}$ and $\beta_{2}$ go to infinity, the optimal bottlenecks $Z_{1}$ and $Z_{2}$ are both deterministic functions of *X* and they do not conflict. When $\beta_{1}=\beta_{2}=0$, the information about *Y* in *X* is maximally compressed in $Z_{1}$ and $Z_{2}$ (i.e., $Z_{1}⊥X,Z_{2}⊥X$), and they do not conflict. The optimal solutions conflict when $\beta_{1}=0$ and $\beta_{2}>0$, as the former leads to a maximally compressed $Z_{1}$ while the latter prefers an informative $Z_{2}$ (this contradicts the Markov structure $X\to Z_{1}\to Z_{2}$, which indicates that maximal compression in $Z_{1}$ leads to maximal compression in $Z_{2}$).

We can also easily construct non-conflicting MIBs for $0<\beta_{1},\beta_{2}<\infty$ that violate the condition. For example, if *X* and *Y* are jointly Gaussian, the optimal bottlenecks $Z_{1}$ and $Z_{2}$ are linear transforms of *X* and jointly Gaussian with *X* and *Y* [11]. In this case, $Z_{2}$ is a sufficient statistic of $Z_{1}$ for *X*. In the case of neural networks, we can also construct a simple but non-trivial neural network that can obtain a non-conflicting Markov information optimality. For example, consider a neural network of two hidden layers $Z_{1}$ and $Z_{2}$, where $Z_{1}$ is arbitrarily mapped from the input layer *X* but $Z_{2}$ is a sample mean of *n* samples i.i.d. drawn from the normal distribution $N(Z_{1};\Sigma )$. This construction guarantees that $Z_{2}$ is a sufficient statistic of $Z_{1}$ for *X*, and thus there is non-conflicting Markov information optimality.

Theorem 1 is a direct result of DPI if $\beta_{i}\in \left\{0,\infty \right\}$. In the case that $0<\beta_{i}<\infty$, we trace down to the Lagrangian multiplier as in the original IB [1] to complete the proof. Formally, before proving Theorem 1, we first establish the two following lemmas. The first lemma expresses the uncertainty reduction in a Markov chain.

**Lemma** **1.**
*Given $Y\to X\to Z_{1}\to Z_{2}$, we have*
(6)$$\begin{array}{cc} I(Z_{2};X) & =I\left(Z_{1};X\right)-I\left(Z_{1};X|Z_{2}\right) \end{array}$$
(7)$$\begin{array}{cc} I(Z_{2};Y) & =I\left(Z_{1};Y\right)-I\left(Z_{1};Y|Z_{2}\right). \end{array}$$


**Proof.** It follows from [27] that $I\left(X;Z_{1};Z_{2}\right)=I\left(X;Z_{2}\right)+I\left(X;Z_{1}|Z_{2}\right)=I\left(X;Z_{1}\right)+I\left(X;Z_{2}|Z_{1}\right)$, but $I(X;Z_{2}|Z_{1})=0$ since 
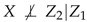
, hence Equation (Equation 6). The proof for Equation (7) is similar by replacing variable *X* with variable *Y*. (Q.E.D.) □

**Lemma** **2.**
*Given $Y\to X\to Z_{1}\to Z_{2},0<\beta_{2}<\infty$ and H(X|Y)>0, let us define the conditional information bottleneck objective:*
(8)$$\begin{array}{c} L^{c}:=L^{c}\left[p\left(z_{2}|z_{1}\right),p\left(z_{1}|x\right)\right]:=I\left(Z_{1};X|Z_{2}\right)-\beta_{2}I\left(Z_{1};Y|Z_{2}\right). \end{array}$$
*If Z is not in the collapse mode, $\partial L^{c}/\partial p\left(z_{1}|x\right)$ depends on $\left\{p\left(z_{2}|z_{1}\right)\right\}$.*


**Proof.** Informally, if $Z_{2}$ in the conditional information bottleneck objective $L^{c}$ is not a trivial transform of the bottleneck variable $Z_{1}$, $Z_{2}$ induces a non-trivial topology into the conditional information bottleneck objective. Formally, by the definition of the conditional mutual information
$$\begin{array}{c} I\left(Z_{1};X|Z_{2}\right)=\int \int \int p\left(x,z_{1},z_{2}\right)\log\frac{p(z_{1},x|z_{2})}{p\left(z_{1}|z_{2}\right)p\left(x|z_{2}\right)}dz_{2}dz_{1}dx, \end{array}$$
$I(Z_{1};X|Z_{2})$ depends on $p(x,z_{1},z_{2})$ as long as the presence of $Z_{2}$ in the conditional information bottleneck objective does not vanish (we will discuss the conditions for $Z_{2}$ to vanish in the final part of this proof). Note that due to the Markov chain $X\to Z_{1}\to Z_{2}$, we have $p\left(x,z_{1},z_{2}\right)=p\left(x\right)p\left(z_{1}|x\right)p\left(z_{2}|z_{1}\right)$.Thus, $\partial I\left(Z_{1};X|Z_{2}\right)/\partial p\left(z_{1}|x\right)$ depends on $p(z_{2}|z_{1})$ as long as $Z_{2}$ does not vanish in the objective. Similarly, the same result also applies to $\partial I\left(Z_{1};Y|Z_{2}\right)/\partial p\left(z_{1}|x\right)$. Hence, $\partial L^{c}/\partial p\left(z_{1}|x\right)$ depends on $\left\{p\left(z_{2}|z_{1}\right)\right\}$ (note that H(X|Y)>0 prevents the collapse of $L^{c}$ when summing two mutual informations) if $Z_{2}$ does not vanish in the objective.Now we discuss the vanishing condition for $Z_{2}$ in the objective. It follows from Lemma 1 that:
(9)$$\begin{array}{c} 0\leq I\left(Z_{1};X|Z_{2}\right)\leq I\left(Z_{1};X\right), \end{array}$$
(10)$$\begin{array}{c} 0\leq I\left(Z_{1};Y|Z_{2}\right)\leq I\left(Z_{1};Y\right). \end{array}$$Note that $Z_{2}$ vanishes in $L^{c}$ iff each of the mutual informations in $L^{c}$ does not depend on $Z_{2}$ iff the equality in both (Equation 9) and (10) occur. If $I(Z_{1};X|Z_{2})=0$, we have $Y\to X\to Z_{2}\to Z_{1}$ (i.e., $Z_{2}$ is a sufficient statistic for *X* and *Y*), which also implies that $I(Z_{1};Y|Z_{2})=0$. Similarly, $I\left(Z_{1};X|Z_{2}\right)=I\left(Z_{1};X\right)$ implies that $Z_{2}$ is independent of $Z_{1}$, which in turn implies that $I\left(Z_{1};Y|Z_{2}\right)=I\left(Z_{1};Y\right)$. (Q.E.D.)  □

We now prove Theorem (1) by using Lemma (Equation 6) and Lemma (7).

**Proof** **of** **Theorem** **1.**(⇐) This direction is obvious. When $I\left(Z_{2};X\right)=I\left(Z_{1};X\right)$ and $I\left(Z_{2};Y\right)=I\left(Z_{1};Y\right)$, or $I(Z_{2};X)=0$ and $I(Z_{2};Y)=0$, there is effectively only one optimization problem for $L_{1}$, and this reduces into the original information bottleneck (with single bottleneck) [1].(⇒) First we prove that if *Z* is not in the collapse mode, the constrained minimization problems are conflicting. Assume, by contradiction, that there exists a solution that minimizes both $L_{1}$ and $L_{2}$ simultaneously, that is, $\exists p\left(z_{1}|x\right),p\left(z_{2}|z_{1}\right)$ s.t. $L_{1}$ has a minimum at $\left\{p\left(z_{1}|x\right)\right\}$ and $L_{2}$ has a minimum at $\left\{p\left(z_{1}|x\right),p\left(z_{2}|z_{1}\right)\right\}$. Note that $\left\{p\left(z_{1}|x\right)\right\}$ and $\left\{p\left(z_{2}|z_{1}\right)\right\}$ are independent variables for the optimization. By introducing Lagrangian multipliers $\lambda_{1}\left(x\right)$ and $\lambda_{2}\left(x\right)$ for the constraint $\int p\left(z_{1}|x\right)dz_{1}=1$ of $L_{1}$ and $L_{2}$, respectively, the stationarity in the Karush–Kuhn–Tucker (KKT) conditions becomes:
(11)$$\begin{array}{c} \frac{\partial L_{1}}{\partial p(z_{1}|x)}=0, \end{array}$$
(12)$$\begin{array}{c} \frac{\partial L_{2}}{\partial p(z_{1}|x)}=0, \end{array}$$
where $L_{1}$ and $L_{2}$ are the Lagrangians:
(13)$$\begin{array}{c} L_{1}\left[p\left(z_{1}|x\right),\lambda_{1}\right]:=I\left(Z_{1};X\right)-\beta_{1}I\left(Z_{1};Y\right)-\int \int \lambda_{1}\left(x\right)p\left(z_{1}|x\right)dz_{1}dx \end{array}$$
(14)$$\begin{array}{c} L_{2}\left[p\left(z_{1}|x\right),\lambda_{2}\right]:=I\left(Z_{2};X\right)-\beta_{2}I\left(Z_{2};Y\right)-\int \int \lambda_{2}\left(x\right)p\left(z_{1}|x\right)dz_{1}dx. \end{array}$$It follows from Lemma 1 that:
(15)$$\begin{array}{c} L_{2}-L_{1}=\left(\beta_{1}-\beta_{2}\right)I\left(Z_{1};Y\right)-L^{c}-\int \int \left(\lambda_{2}\left(x\right)-\lambda_{1}\left(x\right)\right)p\left(z_{1}|x\right)dz_{1}dx, \end{array}$$
where $L^{c}=I\left(Z_{1};X|Z_{2}\right)-\beta_{2}I\left(Z_{1};Y|Z_{2}\right)$ (defined in Lemma 2). We take the derivative w.r.t. $p(z_{1}|x)$ both sides of Equation (Equation 15) and use Equations (Equation 11) and (12):
(16)$$\begin{array}{c} \frac{\partial L^{c}}{\partial p(z_{1}|x)}=\left(\beta_{1}-\beta_{2}\right)\frac{\partial I(Z_{1};Y)}{\partial p(z_{1}|x)}+\lambda_{1}\left(x\right)-\lambda_{2}\left(x\right). \end{array}$$Notice that the left hand side of Equation (Equation 16) strictly depends on $p(z_{2}|z_{1})$ (Lemma 2) while the right hand side is independent of $\left\{p\left(z_{2}|z_{1}\right)\right\}$. This contradiction implies that the initial existence assumption is invalid, and thus implies the conclusion in Theorem 1. (Q.E.D.) □

### 4.1. Markov Information Optimality Compromise

Due to Theorem 1, we cannot simultaneously achieve the information optimality for all bottlenecks. Thus, we need some compromised approach to instead obtain a compromised optimality. We propose two simple compromise strategies, namely, JointMIB and GreedyMIB. JointMIB is a weighted sum of the IB objectives $L^{joint}:=\sum_{l=0}^{L}\gamma_{l}L_{l}$ where $\gamma_{l}\geq 0$. The main idea of JointMIB is to simultaneously optimize all encoders. Even though each bottleneck might not achieve its individual optimality, their joint optimality encourages a joint compromise. On the other hand, GreedyMIB progressively solves the information optimality for each bottleneck given that the encoders for the previous bottlenecks are fixed. In other words, GreedyMIB tries to obtain the conditional optimality of a current bottleneck which is conditioned on the fixed greedy-optimal information of the previous bottlenecks.

### 4.2. Variational Markov Information Bottleneck

Due to the intractability of encoders in Equation (Equation 2) and relevance decoders in Equation (Equation 3), the resulting mutual information in Equation (Equation 4) is also intractable. In this section, we present variational methods to derive a lower bound on mutual information in MIB.

#### 4.2.1. Approximate Relevance

Note that $I\left(Z_{l};Y\right)=H\left(Y\right)-H\left(Y|Z_{l}\right)$, where H(Y)=constant, which can be ignored in the minimization of $L_{l}$. It follows from the non-negativity of KL divergence that:(17)$$\begin{array}{cc} H(Y|Z_{l}) & =-\int p\left(y|z_{l}\right)p\left(z_{l}\right)\log p\left(y|z_{l}\right)dydz_{l}\leq -\int p\left(y|z_{l}\right)p\left(z_{l}\right)\log p_{v}\left(y|z_{l}\right)dydz_{l}\\ \; & \;\;\;=-E_{\left(X,Y\right)}E_{Z_{l}|X}\log p_{v}\left(Y|Z_{l}\right)=-E_{\left(X,Y\right)}E_{Z_{l}|X}\log p\left(\hat{Y}|Z_{l}\right)=:\tilde{H}\left(Y|Z_{l}\right), \end{array}$$
where we specifically use the relevance decoder for surrogate target variable $p_{v}\left(y|z_{l}\right)=p\left(\hat{y}|z_{l}\right)$ as a variational distribution to the intractable distribution $p(y|z_{l})$:(18)$$\begin{array}{cc} p_{v}\left(y|z_{l}\right) & :=E_{Z_{L}|z_{l}}\left[p(\hat{y}|Z_{L})\right]. \end{array}$$

The variational relevance $\tilde{I}\left(Z_{l};Y\right):=H\left(Y\right)-\tilde{H}\left(Y|Z_{l}\right)$ is a lower bound on $I(Z_{l};Y)$. This bound is tightest (i.e., zero gap) when the variational relevance decoder $p(\hat{y}|z_{l})$ equals the relevance decoder $p(y|z_{l})$. In what follows, we establish the relationship between the variational relevance and the log likelihood function, thus connecting MIB with the MLE principle:

**Proposition** **1**(Variational Relevance Inequalities). *Given the definition of variational relevance $\tilde{I}\left(Z_{l};Y\right)=H\left(Y\right)-\tilde{H}\left(Y|Z_{l}\right)$ where $\tilde{H}\left(Y|Z_{l}\right)$ is defined in Equation (Equation 17), and $Z=(Z_{1},\ldots ,Z_{L})$, we have:*
(19)$$\begin{array}{c} H\left(Y\right)+E_{\left(X,Y\right)}[\log p(\hat{Y}|X)]=\tilde{I}\left(Z_{0};Y\right)\geq \tilde{I}\left(Z_{l};Y\right)\geq \tilde{I}\left(Z_{l+1};Y\right)\geq \tilde{I}\left(Z_{L};Y\right)=\tilde{I}\left(Z;Y\right), \end{array}$$
*for all 0≤l≤L−1. where $Z=(Z_{1},\ldots ,Z_{L})$.*


Proposition 1 suggests that: (i) the log likelihood of $p(\hat{y}|x)$ (plus the constant output entropy H(Y)) is a special case of the variational relevance at bottleneck $Z_{0}=X$; (ii) the log likelihood bound $H\left(Y\right)+E_{\left(X,Y\right)}[\log p(\hat{Y}|X)]$ is an upper bound on the variational relevance for all the intermediate bottlenecks $Z_{l}$ and for the composite bottleneck $Z=(Z_{1},\ldots ,Z_{L})$. Therefore, maximizing the log likelihood, as in MLE, does not guarantee to increase the variational relevance for all the the intermediate bottlenecks and the composite bottleneck.

**Proof.** It follows from Jensen’s inequality and the Markov chain that:
$$\begin{array}{cc} \int p\left(z_{l}|x\right)\log p\left(\hat{y}|z_{l}\right)dz_{l} & =\int p\left(z_{l}|x\right)\log\left(\int p\left(\hat{y}|z_{l+1}\right)p\left(z_{l+1}|z_{l}\right)dz_{l+1}\right)dz_{l}\\ \; & \geq \int p\left(z_{l}|x\right)\int p\left(z_{l+1}|z_{l}\right)\log p\left(\hat{y}|z_{l+1}\right)dz_{l+1}dz_{l}\\ \; & =\int \int p\left(z_{l}|x\right)p\left(z_{l+1}|z_{l}\right)\log p\left(\hat{y}|z_{l+1}\right)dz_{l}dz_{l+1}\\ \; & =\int p\left(z_{l+1}|x\right)\log p\left(\hat{y}|z_{l+1}\right)dz_{l+1}, \end{array}$$
for all 0≤l≤L−1. Thus, we have:
$$\begin{array}{cc} \tilde{I}\left(Z_{l};Y\right) & =H\left(Y\right)-\tilde{H}\left(Y|Z_{l}\right)\\ \; & =H\left(Y\right)+E_{\left(X,Y\right)}E_{Z_{l}|X}\log p\left(\hat{Y}|Z_{l}\right)\\ \; & \geq H\left(Y\right)+E_{\left(X,Y\right)}E_{Z_{l+1}|X}\log p\left(\hat{Y}|Z_{l+1}\right)\\ \; & =\tilde{I}\left(Z_{l+1};Y\right). \end{array}$$It also follows from the Markov chain that:
$$\begin{array}{cc} p(\hat{y}|z) & =p\left(\hat{y}|z_{L},z_{L-1},\ldots ,z_{1}\right)=p\left(\hat{y}|z_{L}\right). \end{array}$$Therefore, we have:
$$\begin{array}{cc} \tilde{I}\left(Z;Y\right) & =H\left(Y\right)+E_{\left(X,Y\right)}E_{Z_{L}\left|Z_{L-1},\ldots ,Z_{1}\right|Z_{0}}\log p\left(\hat{Y}|Z_{L}\right)\\ \; & =H\left(Y\right)+E_{\left(X,Y\right)}E_{Z_{L}|X}\log p\left(\hat{Y}|Z_{L}\right)\\ \; & =\tilde{I}\left(Z_{L};Y\right). \end{array}$$Finally, by the definition in Equation (Equation 17), we have:
$$\begin{array}{cc} \tilde{I}\left(Z_{0};Y\right) & =H\left(Y\right)+E_{\left(X,Y\right)}E_{Z_{0}|X}\log p\left(\hat{Y}|Z_{0}\right)\\ \; & =H\left(Y\right)+E_{\left(X,Y\right)}\log p\left(\hat{Y}|X\right). \end{array}$$
(Q.E.D.) □

#### 4.2.2. Approximate Compression

In practice (e.g., in SNN presented in the next section), we can model the encoding between consecutive layers $p(z_{l}|z_{l-1})$ with an analytical form. However, the encoding of non-consecutive layers $p(z_{l}|x)$ for l>1 is generally not analytic as it is a mixture of $p(z_{l}|z_{l-1})$. We thus propose to avoid directly estimating $I(Z_{l};X)$ by instead resorting to its upper bound $I(Z_{l};Z_{l-1})$ as its surrogate in the optimization. However, $I(Z_{l};Z_{l-1})$ is still intractable as it involves the intractable marginal distribution $p\left(z_{l}\right)=\int p\left(z_{l}|x\right)p\left(x\right)dx$. We then approximate $I(Z_{l};Z_{l-1})$ using a mean-field (factorized) variational distribution $q\left(z_{l}\right)=\prod_{i=1}^{n_{l}}q\left(z_{l,i}\right)$ where $z_{l}=\left(z_{l,1},\ldots ,z_{l,n_{l}}\right)$:
(20)$$\begin{array}{cc} I(Z_{l};X) & \leq I\left(Z_{l};Z_{l-1}\right)=\int p\left(z_{l}|z_{l-1}\right)p\left(z_{l-1}\right)\log\frac{p(z_{l}|z_{l-1})}{p(z_{l})}dz_{l}dz_{l-1}\\ \; & \leq \int p\left(z_{l}|z_{l-1}\right)p\left(z_{l-1}\right)\log\frac{p(z_{l}|z_{l-1})}{q(z_{l})}dz_{l}dz_{l-1}=E_{Z_{l-1}}D_{KL}\left[p\left(Z_{l}|Z_{l-1}\right)\left|\right|q\left(Z_{l}\right)\right]\\ \; & =E_{Z_{l-1}}\sum_{i=1}^{n_{l}}D_{KL}\left[p\left(Z_{l,i}|Z_{l-1}\right)\left|\right|q\left(Z_{l,i}\right)\right]=:\tilde{I}\left(Z_{l};Z_{l-1}\right). \end{array}$$

The mean-field variational inference not only helps derive a tractable approximation but also encourages distributed representation by constraining each neuron to capture an independent factor of variation for the data [32]; thus, it can potentially represent an exponential number of concepts using independent factors.

## 5. Case Study: Learning Binary Stochastic Neural Networks

In this section, we officially connect the variational MIB in Section 4 to stochastic neural networks (SNNs). We consider an SNN with *L* hidden layers (without any feedback or skip connection) where the input layer *X*, the hidden layers $Z_{l}$ for 1≤l≤L, and the output layer $\hat{Y}$ are considered as random variables. We use the convention that $Z_{0}:=X,Z_{L+1}:=\hat{Y},$ and $Z_{l}=\emptyset$ for all l∉{0,1,…,L,L+1}. Without any feedback or skip connection, $Y,X,Z_{l},Z_{l+1},$ and $\hat{Y}$ form a Markov chain in that order. The output layer $\hat{Y}$ is the surrogate target variable presented in Section 4. The role of SNNs is therefore reduced to transforming from one random variable to another via the Markov chain $X\to Z_{l}\to Z_{l+1}\to \hat{Y}$ such that it achieves the good information representation (i.e., the compression–relevance tradeoff) for each layer. With the MLE principle, the learning in SNNs is performed by maximizing the log likelihood $E_{\left(X,Y\right)}[\log p(\hat{Y}|X)]$. However, maximizing the log likelihood does not guarantee to improve the variational relevance for all the intermediate bottlenecks and the composite bottleneck (Proposition 1).    



**Algorithm 1:**
JointMIB


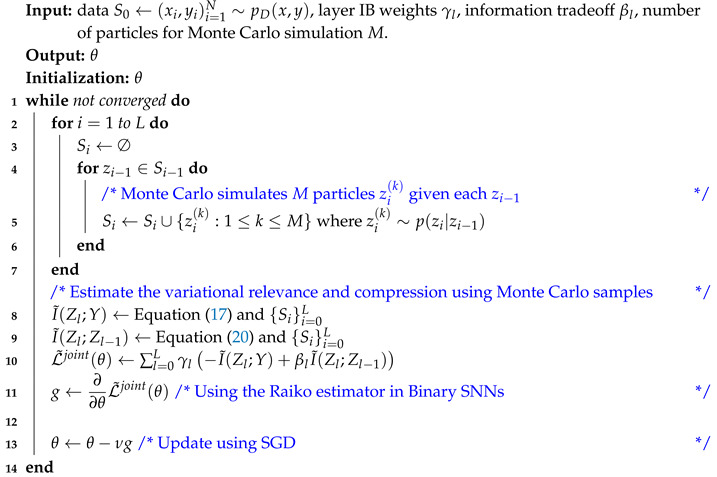




We here instead combine the variational MIB and the MIB compromise to derive a practical learning principle that encourages compression and relevance for each layer, improving the information flow in SNNs. To make it concrete and simple, we consider a simple network architecture: binary stochastic feed-forward (fully-connected) neural networks (SFNNs). In binary SFNNs, we use a sigmoid as the activation function: $p\left(z_{l}=1|z_{l-1}\right)=\sigma \left(W_{l-1}z_{l-1}+b_{l-1}\right)$, where σ(.) is the (element-wise) sigmoid function, $W_{l-1}$ is the network weights connecting layer l−1 to layer *l*, $b_{l-1}$ is a bias vector, and $Z_{l}\in {\left\{0,1\right\}}^{n_{l}}$. Let us define ${\tilde{L}}_{l}:=-\tilde{I}\left(Z_{l};Y\right)+\beta_{l}\tilde{I}\left(Z_{l};Z_{l-1}\right)$, where $\tilde{I}\left(Z_{l};Y\right)$ and $\tilde{I}\left(Z_{l};Z_{l-1}\right)$ are the approximate relevance and compression defined in Equation (Equation 17) and (Equation 20), respectively. Note that the position of $\beta_{l}$ here is slightly different from its position in Equation (Equation 4). In Equation (Equation 4), $\beta_{l}$ is associated with the relevance term to respect the convention of the original IB, while here it is associated with the compression term for practical reasons. In practice, the contribution of $\tilde{I}\left(Z_{l};Y\right)$ is higher than $\tilde{I}\left(Z_{l};Z_{l-1}\right)$. In computing $\tilde{I}\left(Z_{l};Y\right)$ and $\tilde{I}\left(Z_{l};Z_{l-1}\right)$, any expectation with respect to $p(z_{l}|z_{l-1})$ is approximated by Monte Carlo simulation in which we sample *M* particles $z_{l}∼p\left(z_{l}|z_{l-1}\right)$. Regarding the information optimality compromise, we combine the variational MIB objectives into a weighted sum in JointMIB:(21)$$\begin{array}{c} {\tilde{L}}^{joint}:=\sum_{l=0}^{L}\gamma_{l}{\tilde{L}}_{l}, \end{array}$$
where $\gamma_{l}\geq 0$. In GreedyMIB, we greedily minimize ${\tilde{L}}_{l}$ for each 0≤l≤L. We also make each $q(Z_{l,i})$ a learnable Bernoulli distribution. The JointMIB is presented in Algorithm 1. The Monte Carlo sampling operation of Algorithm 1 in stochastic neural networks precludes the backpropagation in a computation graph. It becomes even more challenging with binary stochastic neural networks, as it is not well-defined to compute gradients w.r.t. discrete-valued variables. Fortunately, we can find approximate gradients, which have been proved to be efficient in practice: the REINFORCE estimator [33,34], the straight-through estimator [35], the generalized EM algorithm [20], and the Raiko (biased) estimator [21]. Especially, we found that the Raiko gradient estimator works best in our specific setting and thus deployed it in this application. In the Raiko estimator, the gradient of a bottleneck particle $z_{l,i}∼p\left(z_{l,i}=1|z_{l-1}\right)=\sigma \left(a_{i}^{\left(l\right)}\right)$ is propagated only through the deterministic term $\sigma (a_{i}^{\left(l\right)})$: $\frac{\partial z_{l,i}}{\partial \theta }\approx \frac{\partial \sigma (a_{i}^{\left(l\right)})}{\partial \theta }.$

## 6. Experimental Evaluation

We evaluated the effectiveness of the MIB framework on binary SNNs in synthetic data and MNIST hand-digit recognition data [23]. Each data sample in MNIST is a 28×28 gray-scale image representing a handwritten digit from 0 to 9. The dataset is split into 60000 training samples and 1000 test samples. In the synthetic data, we visualized the learning dynamics of the SNNs trained with the variational MIB variants (i.e., JointMIB and GreedyMIB), and those trained with MLE. In MNIST, we evaluate the effectiveness of the variational MIB variants by comparing them against the baselines MLE and VIB [14] in classification, adversarial robustness and multi-modal learning problems. We make the code for our framework publicly available at https://github.com/thanhnguyentang/pib.

### 6.1. Synthetic Data: Learning Dynamics of Variational MIB

To better understand how MIB modified the information within the layers during the learning process, we visualized the compression and relevance of each layer over the course of training of stochastic feed-forward neural networks (SFNNs) [21], JointMIB, and GreedyMIB in synthetic data. SFN1N is different from MIB only in the objective functions: SFNN is trained with the negative log likelihood while MIB is trained with the variational MIB objective. To simplify our analysis, we considered a binary decision problem where *X* is 12 binary inputs making up $2^{12}=4096$ equally likely input patterns and *Y* is a binary variable equally distributed among 4096 input patterns [16]. The base neural network architecture had 4 hidden layers with widths: 10–8–6–4 neurons. Since the network architecture was small, we could precisely compute the true compression $I_{x}:=I\left(Z_{i};X\right)$ and true relevance $I_{y}:=I\left(Z_{i};Y\right)$ over training epochs. We fixed $\beta_{l}=\beta =10^{-4}$ for both JointMIB, trained five different randomly initialized neural networks for each comparative model with stochastic gradient descent (SGD) up to 20,000 epochs on 80% of the data, and averaged the mutual information. In JointMIB, we set $\gamma_{l}=\gamma =1,\forall l$.

Figure 2 provides a visualization of the learning dynamics of SFNN versus JointMIB on the information plane $\left(I_{x},I_{y}\right)$. Firstly, we observed a common trend in the learning dynamics of MLE (in the SFNN model) and JointMIB frameworks. Both principles allow the network to gradually encode more information about *X* and the relevant information about *Y* into the hidden layers at the beginning as $I(Z_{i};X)$ and $I(Z_{i};Y)$ both increase. Intuitively, in order for the representations $Z_{l}$ to make sense of the task, the representations should encode enough information about *X*; thus, $I(Z_{l};X)$ should increase. This is especially true for shallow layers because, due to the Markov chain property, the shallower a layer, the greater its burden of carrying enough information to make sense of a task. Especially, we can observe that the increase of $I(Z_{l};X)$ slowed down at some point for the deeper layers for both SFNN and MIB. This slowing effect was especially stronger in MIB where the compression is explicitly encouraged during the learning. Secondly, MIB was different from MLE in the maximum level of relevance at each layer and the number of epochs to encode the same level of relevance. JointMIB at l=1 needed only about 4.68% of the training epochs to achieve at least the same level of relevance in all layers of SFNN at the final epoch. In addition, MLE was unable to encode the network layers to reach the maximum level of relevance enabled by MIB (we also trained SFNN up to 100,000 epochs and observed that the level of relevance of each layer never reached the value of 0.8 bits).

There is also a subtle observation in Figure 2 that the relevance for MIB increased until some point before decreasing, while the relevance for SFNN increased until some point where the value almost stayed the same without a noticeable decrease. This could be explained by the fact that that the MIB objective can eventually allow the encoding of relevant information into each layer to its optimal information trade-off at some point. After this point, if training is continued, due to the mismatch between the exact MIB objective and its variational bound, the further minimization of the variational bound would decrease $I(Z_{l};Y)$. Consequently, in order for $\beta_{l}I\left(Z_{l};X\right)-I\left(Z_{l};Y\right)$ to be small, $I(Z_{l};X)$ also needs to decrease after this point to compensate for the decrease in $I(Z_{l};Y)$. In the case of SFNN (trained with MLE), the MLE objective reaches its local minimum before the information of each layer can even reach its optimal information trade-off (if ever). This also suggests that MIB is better than MLE in terms of exploiting information for each layer during the learning.

GreedyMIB also obtained the representation of higher relevance as compared to MLE (Figure 3). GreedyMIB at l=1 needed only about 17.95% of the training epochs to achieve at least the same level of relevance in all layers of the SFNN at the final epoch. Recall that in GreedyMIB at l=1 the MIB principle is applied only to the first hidden layer. The layer representation at the final epoch gradually shifts to the left (i.e., more compressed) while not degrading the relevance over the greedy training from layer 1 to layer 4 in Figure 3.

We also see the compression effect that the compression constraints within the MIB framework prevented the layer representation from shifting to the right (in the information plane) during the encoding of relevant information (e.g., it slowed down the increase of $I(Z_{l};X)$ during the information encoding, keeping the representation more concise). As compared with JointMIB, GreedyMIB also obtained a comparable information representation.

To conclude, two main advantages of MIB as compared to MLE are: (i) MIB can improve the information representation in SNNs in terms of higher relevance while keeping the information in each layer concise during encoding; (ii) MIB uses much fewer training epochs to obtain such information representation.

### 6.2. Image Classification

In this experiment, we compared JointMIB and GreedyMIB with three other comparative models which used the same network architecture without any explicit regularizer: (1) a standard deterministic neural network (DET) which simply treated each hidden layer as deterministic; (2) a stochastic feed-forward neural network (SFNN) [21] which is a binary stochastic neural network as in MIB but is trained with the MLE principle; and (3) variational information bottleneck (VIB) [14], which uses the entire deterministic network as an encoder, adds an extra stochastic layer as a out-of-network bottleneck variable, and is then trained with the IB principle on that single bottleneck layer. The base network architecture in this experiment had two hidden layers with 512 sigmoid-activated neurons per layer. These models were trained in MNIST [23].

Adopted from the common practice, we used the last 10,000 images of the training set as a validation (holdout) set for tuning hyperparameters. We then retrained the models from scratch in the full training set with the best validated configuration. We trained each of the five models with the same set of five different initializations and reported the average results over the set. For the stochastic models (all except DET), we drew M=32 samples per stochastic layer during both training and inference, and performed inference 10 times at test time to report the mean classification errors for MNIST. The value of M=32 is empirically reasonable in this experiment, as illustrated in Figure 4.

For JointMIB and GreedyMIB, we set $\gamma_{l}=1$ (in JointMIB only) and $\beta_{l}=\beta ,\forall 1\leq l\leq L$, tuned β on a linear log scale $\beta \in \{10^{-i}:1\leq i\leq 10\}$. We found $\beta =10^{-4}$ worked best for both models (Figure 5). For VIB, we found that $\beta =10^{-3}$ worked best on MNIST. We trained all the models on MNIST with Adadelta optimization [36], except for VIB for which we used Adam optimization [37], as we found that they worked best in the validation set.

The results are shown in Table 1. It shows that JointMIB substantially outperformed DET, MLE, and VIB on MNIST while GreedyMIB outperformed only DET and underperformed SFNN. Though JointMIB and GreedyMIB could have comparable information representation, as illustrated in the synthetic experiment in Section 6.1, in practice, it can be harder to obtain a comparable information representation for GreedyMIB. In GreedyMIB, it is necessary to train each layer greedily in order to obtain its information representation. The greedy nature makes it difficult to determine when would be a good time to stop the training and conclude the information representation for each layer. In addition, training greedily is expensive. JointMIB makes it more efficient by jointly obtaining a compromised information representation in each layer. Thus, it allows the compromised information representations of all the layers to jointly interact with each other during the learning. In principle, it is also harder to obtain good information representation in GreedyMIB. Due to the conflicting information optimality in MIB (Theorem 1), the good encoder for the first layer does not guarantee a good information trade-off in the the deeper layers. Though JointMIB also suffers from the conflicting information optimality, jointly and explicitly inducing relevant but compressed information into each layer of a neural network via MIBs as in JointMIB can make it easier for the training.

### 6.3. Robustness against Adversarial Attacks

We consider here the adversarial robustness of neural networks trained with MIBs. Neural networks are prone to adversarial attacks which disturb the input pixels by small amounts that are imperceptible to humans [38,39]. Adversarial attacks generally fall into two categories: untargeted and targeted attacks. An untargeted adversarial attack A maps the target model *M* and an input image *x* into an adversarially perturbed image $x^{\prime }$: $A:\left(M,x\right)\to x^{\prime }$, and is considered successful if it can fool the model $M\left(x\right)\neq M\left(x^{\prime }\right)$. A targeted attack, on the other hand, has an additional target label *l*: $A:\left(M,x,l\right)\to x^{\prime }$, and is considered successful if $M\left(x^{\prime }\right)=l\neq M\left(x\right)$.

We performed adversarial attacks on the neural networks trained with MLE and MIB, and used the accuracy on adversarially perturbed versions of the test set to rank a model’s robustness. In addition, we used the $L_{2}$ attack method for both targeted and untargeted attacks [40], which has shown to be the most effective attack algorithm with smaller perturbations. Specifically, we attacked the same four comparative models described from the previous experiment on the first 1000 samples of the MNIST test set. For the targeted attacks, we targeted each image into the other 9 labels other than the true label of the image. We used the same hyperparameters as in the classification experiment. The value of $\beta =\beta_{l}=10^{-4}$ was also reasonable for this adversarial robustness task (Figure 6).

The results are shown in Table 1. Firstly, it was expected that the adversarial robustness accuracy in the targeted attacks would be smaller than that in the untargeted attacks because the targeted attacks are more challenging for the neural networks to overcome than untargeted attacks. This result is consistent in our experiment. Secondly, the deterministic model DET was totally fooled by all the attacks. It is known that stochasticity in neural networks improves adversarial robustness, which is consistent with our experiment as SFNN was significantly more adversarially robust than DET. Thirdly, VIB had comparable adversarial robustness to SFNN even if VIB had “less stochasticity” than SFNN (VIB had one stochastic layer while all hidden layers of the SFNN were stochastic). We hypothesize that this is because VIB performance was compensated with the IB principle for its stochastic layer. Finally, JointMIB was more adversarially robust than the other models. Again, GreedyMIB was not very effective in adversarial robustness (it was worse than VIB in the targeted attack and SFNN in the untargeted attack). We hypothesize that this relates to the difficulty for GreedyMIB to have a good information representation for all layers. In conclusion, this experiment suggests that explicitly and jointly inducing compression and relevance into each layer has a good potential of being more adversarially robust for neural networks.

### 6.4. Multi-Modal Learning

One of the main advantages of stochastic neural networks is their ability to model structured output space in which a one-to-many mapping is required. A binary stochastic variable $z_{l}$ of dimensionality $n_{l}$ can take on $2^{n_{l}}$ different states, each of which would give a different $\hat{y}$. Thus, the conditional distribution $p(\hat{y}|x)$ in stochastic neural networks is multi-modal. Hence in this experiment, we evaluated how MIB affected the multi-modal learning capability of SNNs.

In this experiment, we followed [21] and predicted the lower half of the MNIST digits using the upper half as inputs. We used the same neural network architecture of 392–512–512–392 for JointMIB and SFNN and trained them with SGD with a constant learning rate of 0.01 (due to the under-performance of GreedyMIB from the previous experiments and its expensive training, we compared only JointMIB with SFNN in this experiment). We trained the models on the full training set of 60,000 images and tested with the test set. For JointMIB, we also used $\beta_{l}=\beta =10^{-4}$. The results of JointMIB at epoch 60 and MLE at epoch 200 are shown in Figure 7. Firstly, JointMIB could generate digit variations which were more recognizable than those generated by MLE. In particular, some samples of digits 2, 4, 5, and 7 generated by MLE were distorted, while all digit samples generated by JointMIB were recognizable. Secondly, JointMIB used much fewer epochs to achieve good samples. In JointMIB, we trained only up to 60 epochs while in MLE, we trained up to 200 epochs but did not observe as good samples in between. This further highlights the advantage of MIB in obtaining good information representation in much fewer training epochs. Furthermore, we expect that the advantage of inducing compression and relevance into each layer by JointMIB is particularly helpful for multi-modal learning because in multi-modal learning, the modes generated in each hidden layer are critical for representing multiple modes. While MLE ignores the explicit contribution of each layer to the information representation of the neural network, JointMIB explicitly takes into account the compression and relevance of each layer.

## 7. Discussion and Future Work

In this work, we introduce Markov Information Bottleneck, an extension of the original Information Bottleneck to the context where a representation is multiple stochastic variables that form a Markov chain. In this context, we show that one cannot simply directly apply the original IB principle to each variable as their information optimality is conflicting for most of the interesting cases. We suggest a simple but efficient fix via a joint compromise. In this scheme, we jointly combine the information trade-offs of each variable into a weighted sum, encouraging the information trade-offs for all the variables better off during the learning. In particular in the context of Stochastic Neural Networks, we present the variational inference to estimate the compression and relevance for each bottleneck. As a result, the variational MIB turns the intractable decoding of each bottleneck approximately into an efficient inference for that bottleneck. This variational approximation turns out to generalize the MLE principle in the context of Stochastic Neural Networks. We empirically demonstrate the effectiveness of MIB by comparing it with the baselines using MLE principle and Variational Information Bottleneck in classification, adversarial robustness and multi-modal learning. The empirical performance supports the potential benefit of explicitly inducing compression and relevance into each layer (e.g., in a jointly manner), presenting a special link between information representation and the performance in classification, adversarial robustness and multi-modal learning.

One limitation of our current approach is the number of samples generated via $z_{l}∼p\left(z_{l}|z_{l-1}\right)$ used to estimate the variational compression and relevance scales exponentially with the number of layers. This is however a common drawback for performing inference in fully stochastic neural networks. This difficulty can be overcome by using partially stochastic neural networks. In addition, the Monte Carlo sampling to estimate the variational mutual information, though unbiased, is of high variance and sample inefficiency. This sample inefficiency limitation can be overcome by resorting to more advanced methods of estimating mutual information such as [41,42]. The MIB framework also admits several possible future extensions including scaling the framework to bigger networks and real-valued stochastic neural networks. The extension to real-valued stochastic neural networks are straightforward by, e.g., constructing a Gaussian layer for modeling $p(z_{l}|z_{l-1})$ and using reparameterization tricks [43] to perform back-propagation via sampling. Another dimension of improvement is to study hyperparameter effect of MIB. This current work only considers equal $\gamma_{l}=\gamma$ for JointMIB and equal $\beta_{l}=\beta$, and tuned β via grid search. We can use, e.g., Bayesian optimization [44] to efficiently tune $\gamma_{l}$ and $\beta_{l}$ with expandable bounds. In addition, we believe that the challenge of applying our methods to more advanced datasets such as Imagenet [45] is partly associated with that of scaling the stochastic neural network as we tend to need more expressive models for more challenging datasets. Given this perspective, the challenge to scale to large datasets can be partially addressed with the solutions from scaling stochastic neural networks some of which we suggest above. Furthermore, we believe that, as one of the main messages from our work, explicitly inducing compressed and relevant information (e.g., via mutual information as in MIB) into many intermediate layers can be more beneficial to large-scale tasks than simply resorting to the MLE principle. An intuition is to think of this as a way for *information-theoretic regularization for intermediate layers*. Finally, a followup important question to ask is whether there is any theoretical and stronger empirical link between an improved information representation (e.g., in the MIB sense) and the generalization of neural networks. This connection might be intuitively correct but a systematically empirical study or a theoretical suggestion are an important future research direction.

## Figures and Tables

**Figure 1 entropy-21-00976-f001:**
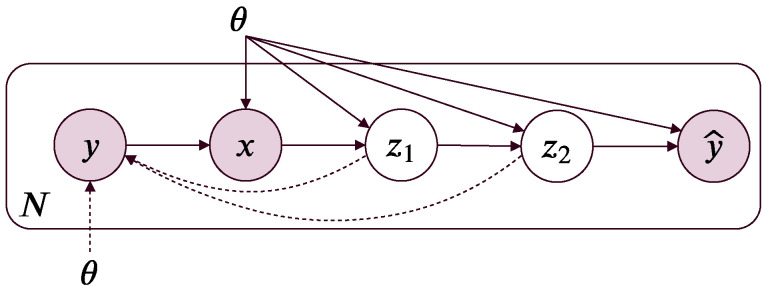
A directed graphical model for the Markov information bottleneck of two Markov bottlenecks. In a non-collapse Markov chain $Y\to X\to Z_{1}\to Z_{2}$ with $0<\beta_{1},\beta_{2}<\infty$, the information optimality in $Z_{1}$ prevents the information optimality in $Z_{2}$. Solid lines denote the encoders $p_{\theta }\left(z_{i}|x\right)$ (for i∈{1,2}), dashed lines denote the variational approximations $p_{\theta }\left(\hat{y}|z_{i}\right)$ to the intractable *relevance* decoder $p_{\theta }\left(y|z_{i}\right)$. The variational relevance decoder $p_{\theta }\left(\hat{y}|z_{i}\right)$ encodes the information from $z_{i}$ into a surrogate target variable $\hat{y}$. In the case of stochastic neural networks, $z_{i}$, θ, and the surrogate target variable represents the hidden layers, the network weights, and the output layer, respectively.

**Figure 2 entropy-21-00976-f002:**
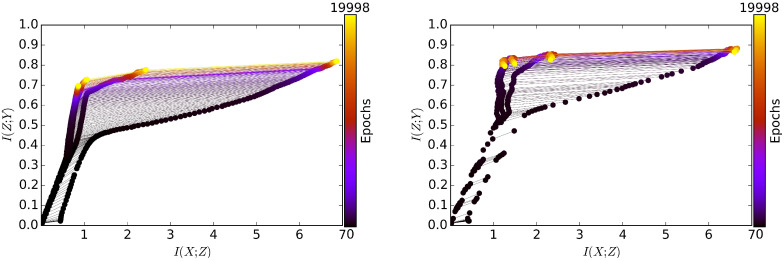
The learning dynamics of the stochastic feed-forward (fully-connected) neural network (SFNN) (**left**) and JointMIB (**right**). The color indicates the training epochs while each node in a color in the graph represents $\left(I\left(Z_{l};X\right),I\left(Z_{l};Y\right)\right)$ at the corresponding epoch. Note that at each epoch, $I\left(Z_{l};X\right)\geq I\left(Z_{l+1};X\right),\forall l$ (data processing inequality—DPI). JointMIB jointly encodes relevant information into every layer of stochastic neural networks (SNNs), while keeping each layer informatively concise. Compared to maximum likelihood estimation (MLE), the level of relevant information encoded by JointMIB increased more quickly over training epochs and reached a higher value. MIB: Markov information bottleneck.

**Figure 3 entropy-21-00976-f003:**
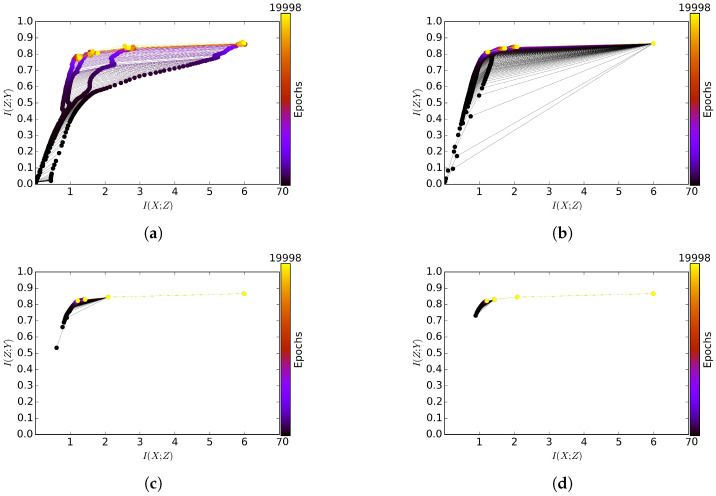
Subfigures (**a**–**d**) represent GreedyMIB’s encoding of relevant information into layers 1 ≤ l ≤ 4, respectively. GreedyMIB greedily encodes relevant information into each layer given the encoded information of the previous layers. GreedyMIB also achieved a significantly higher level of relevant information at each layer compared to MLE

**Figure 4 entropy-21-00976-f004:**
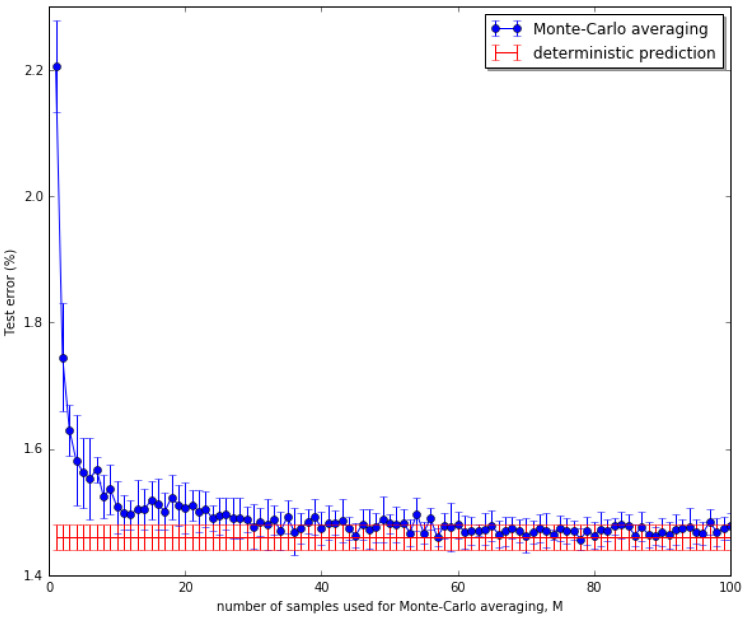
The value of *M* versus validation error. M=32 gave a reasonably good performance as compared to other larger values.

**Figure 5 entropy-21-00976-f005:**
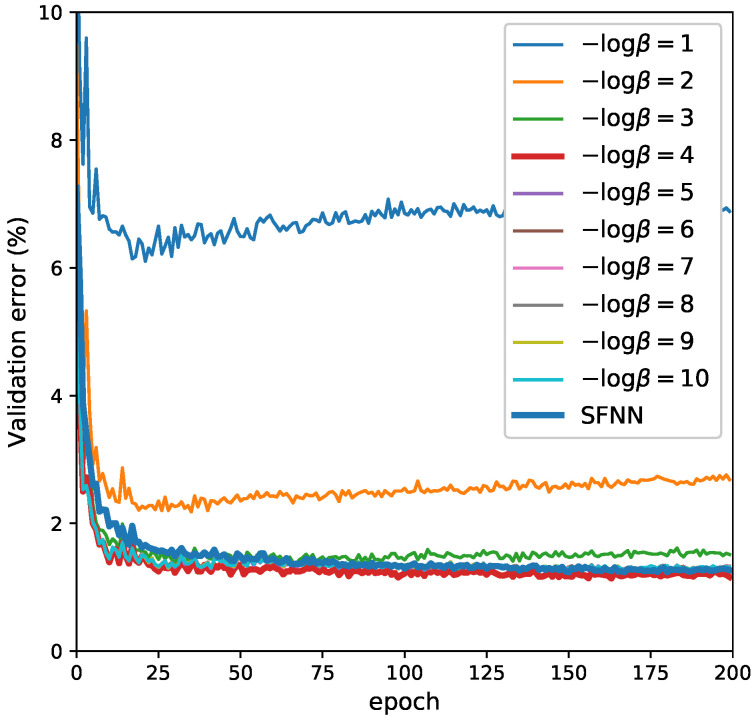
The learning curve of JointMIB and SFNN in the MNIST validation set. Either a too-large or too-small value of β could hurt the generalization of learning. While a large value of β introduces aggressive compression, a smaller value allows more irrelevant information into the representation. In this experiment, we found that $\beta =10^{-4}$ was the best trade-off hyperparameter in JointMIB for this experiment.

**Figure 6 entropy-21-00976-f006:**
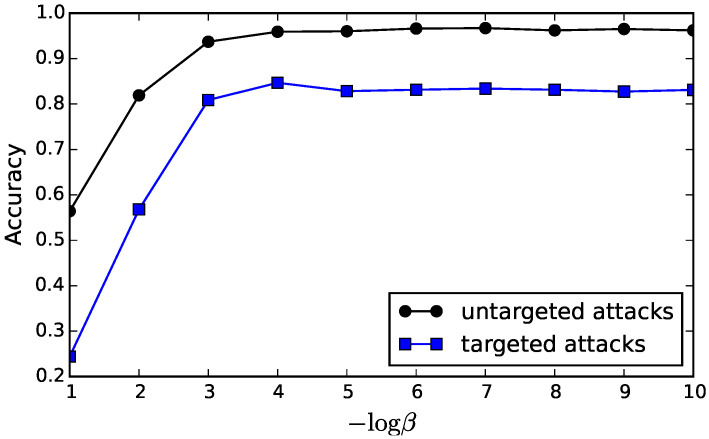
Adversarial robustness of JointMIB for various values of β. Introducing aggressive compression (i.e., large values of β) reduced adversarial robustness while smaller values of β introduced comparable robustness. The best information trade-off for targeted attacks was at $\beta =10^{-4}$ in this experiment.

**Figure 7 entropy-21-00976-f007:**
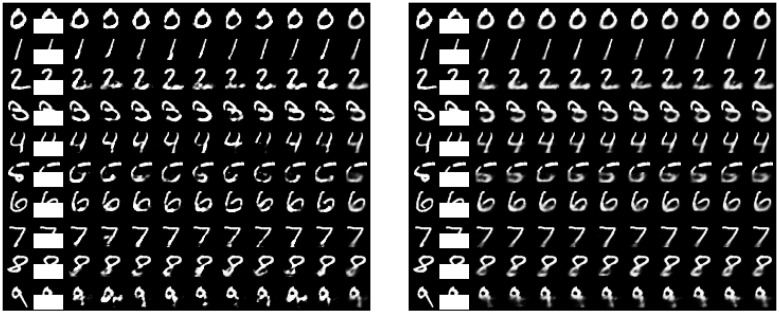
Samples drawn from the prediction of the lower half of the MNIST test data digits based on the upper half for JointMIB (**right**, after 60 epochs) and SFNN (**left**, after 200 epochs). The leftmost column is the original MNIST test digit followed by the masked out digits and nine samples. The rightmost column was obtained by averaging over all generated samples of bottlenecks drawn from the prediction. The figures illustrate the capability of modeling structured output space using JointMIB and SFNN. JointMIB generated more recognizable digits within much fewer training epochs.

**Table 1 entropy-21-00976-t001:** The performance of the variational MIB variants (i.e., JointMIB and GreedyMIB) for classification and adversarial robustness on MNIST in comparison with MLE and variational information bottleneck (VIB). MIB explicitly induces compression–relevance trade-offs in each layer during the training, which outperforms and is more adversarially robust than the other models of the same architecture. DET: deterministic neural network.

Model	Classification MNIST (Error %)	Adv. Robustness (%)	
		Targeted	Untargeted
DET	1.73	00.00	00.00
VIB [14]	1.45	83.70	93.10
SFNN [21]	1.44	83.00	95.20
GreedyMIB	1.54	83.21	94.30
JointMIB	1.36	84.16	96.00

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
