# Peer review of "Markov Information Bottleneck to Improve Information Flow in Stochastic Neural Networks"

_entropy, 2019, doi:10.3390/e21100976_

Round 1

Reviewer 1 Report

The results are interesting. Do you have results of image classification and multimodal learning in a database more recent and complex than MNIST? Some other existing methods report lower classification rates on MNIST dataset than those presented on Table 1. Can you explain please. Can you try in a dataset of faces? 2. There are several of your references from Conferences and arXiv papers.  Can you change some of these references by Journal references. 

Reviewer 2 Report

General assessment

This eloquently written manuscript fits into machine learning theorization. In fact, it is about dealing with Markov Information Bottleneck and applied into stochastic neural networks context. Such theme is current if one considers the possible outcomes and development of Authors’ suggestion. The manuscript is adequate for this Journal’s scope.

There is substance both in the hypothesis and on the experimental investigation. The problem is sufficiently defined, yet, the shortcomings it aims to address could be further explained and exemplified.

Methods are sufficiently explained, yet dispersed along with the manuscript. In fact, a twofold approach has been undertaken for research methods. For the hypothesis statement and formal assessment, it is adequate. On the other hand, for the experimental assessment, methods do not comprise systematic and extensive studies. Instead, those are simply two examples.

However, in order to make it fully replicable, I advise Authors to cite experimental data sources. Furthermore, Authors’ preprint “Information Multi-Bottlenecks” and first Author Master thesis “Parametric Information Bottleneck to OptimizeStochastic Neural Networks” could be cited and dicussed, given the fact that not so impressive, but not less interesting, results have been attained for different examples.

Conclusions, while supported by the results, are neither profuse nor clear. In fact, there is not an explicit reflexion for drawing conclusions. Being the least developed part of a good research article, I suggest improving it.

While regarded as a rather small and prospective development, the content of this manuscript is innovative and may lead to important future developments.

The manuscript structure clearly falls out of orthodoxy, which may hinder its intelligibility. Sections such as “Preliminaries” and merged “Discussion and Future work” without a synthesized conclusions section illustrate my statement. This, however, may be considered acceptable, given the mathematical ground of much of the work. On the other hand, structural singularities, such as having long, not sub-chapterized, blocks of text between, for example, 4 and 4.1, is not acceptable. Please consider re-structuring.

All things considered, this is a good research article, which deserves to be published, provided some minor issues are addressed.

Further, specific comments

Despite these vast applications and variants of IB, alongside with theoretical analysis of the IB principle in neural networks [15,16], it is not clear how the IB insight can be leveraged to improve the learning of stochastic neural networks where hidden layers can be considered as random variables that form a Markov chain.”

From my point of view, this statement lacks support from bibliography.

 “We then empirically show that MIB improves the performance in classification, adversarial robust learning and multi-modal learning in MNIST.

Please consider explaining the meaning of MNIST dataset.

About English writing

In “Stochastic neural networks (SNNs) is a” please consider “Stochastic neural networks (SNNs) are a”

In “One of advantages of SNNs” please consider “One of the advantages of SNNs”

In “surrogate target variable represent” please consider “surrogate target variable represents”

which we will present in detail in next section please consider “which we will present in detail in the next section”

In “are both deterministic function of” please consider “are both a deterministic function of”

In “we can find 211 approximate gradients which has been proved to be efficient” please consider “we can find 211 approximate gradients which have been proved to be efficient”

In “which has shown to be most effective attack algorithm” please consider “which has shown to be the most effective attack algorithm”

In “reduces adversarial robustness while smaller values of β introduces comparable robustness.” please consider “reduces adversarial robustness while smaller values of β introduce comparable robustness.”

In “for JointMIB and SFNN and trained them with SGD with constant learning rate” please consider “for JointMIB and SFNN and trained them with SGD with a constant learning rate”

In “This work was supported by Institute for Information and Communications Technology Planning” “This work was supported by the Institute for Information and Communications Technology Planning”

In “Data Processing Inequalty” please consider “Data Processing Inequality”

In “trade-off in the the deeper layers” please consider “trade-off in the deeper layers”

Change “ouput” to “output”.

Reviewer 3 Report

Please:

elaborate on eq. 1 (transition from Lagrangian to its approximation) describe how the proposed compressed scheme address important corner cases (e.g. main challenge in autonomous diving) which are not well reflected in statistics of the input data, elaborate on number of bit- line 141, could you make the code used for experiments available to the community (e.g. github), how about a hybrid approach (line 177) in addition to JointMIB and Greedy MIB, compare your approach with other Deep Learning compression schemes, explain a challenge of using more advanced datasets (e.g. imagenet) with your flow.